# Recent Developments in NSG and NRG Humanized Mouse Models for Their Use in Viral and Immune Research

**DOI:** 10.3390/v15020478

**Published:** 2023-02-09

**Authors:** Maksym Kitsera, Jesús Emanuel Brunetti, Estefanía Rodríguez

**Affiliations:** 1Bernhard-Nocht Institute for Tropical Medicine, 20359 Hamburg, Germany; 2German Center for Infection Research, Partner Site Hamburg-Borstel-Lübeck, 22297 Hamburg, Germany

**Keywords:** humanized mice, SCID, NOD/SCID, NSG, NRG, viruses, infection

## Abstract

Humanized mouse models have been widely used in virology, immunology, and oncology in the last decade. With advances in the generation of knockout mouse strains, it is now possible to generate animals in which human immune cells or human tissue can be engrafted. These models have been used for the study of human infectious diseases, cancers, and autoimmune diseases. In recent years, there has been an increase in the use of humanized mice to model human-specific viral infections. A human immune system in these models is crucial to understand the pathogenesis observed in human patients, which allows for better treatment design and vaccine development. Recent advances in our knowledge about viral pathogenicity and immune response using NSG and NRG mice are reviewed in this paper.

## 1. Introduction

The use of immunocompetent animals in research has been beneficial to a variety of scientific disciplines. In the case of mice, their genetics and immune system differ significantly from those in humans, which is a challenge to study human-specific diseases in common laboratory mouse strains [1]. This is because some viruses that cause human disease are species-specific and cannot infect mice. Others do it, but the diseases they cause do not entirely resemble the diseases observed in humans. Hence, the generation of mice carrying human immune systems has helped to address these species-specific differences [2]. In order to achieve mouse humanization, it has been crucial to develop mouse models that can support the engraftment of human hematopoietic stem cells, to replicate the complexity of human hematopoiesis. Moreover, the human immune systems in these models can mount immune responses that are both functional and spatially similar to the ones in humans.

The first mouse model with severe combined immunodeficiency (SCID) was created in the early 1980s [3]. This mouse was generated in the CB17 strain and carried the mutation Prkdc^scid^ (protein kinase, DNA activated, catalytic polypeptide), which allowed the engraftment of human immune cells, such as peripheral blood mononuclear cells (PBMCs) [4] and hematopoietic stem cells (HSCs), as well as human tissue [5]. However, due to the spontaneous generation of mouse T-, B-, and NK cells during aging, this model had a limited capacity to efficiently and long-term engraft human cells [6]. Another significant step toward generating mouse strains suitable for humanization was done with the development of immunodeficient non-obese diabetic (NOD) mice. This mutation was combined with the SCID mutation in the mid-1990s [7]. Compared to SCID mice, NOD-SCID mice showed higher engraftment levels of human HSCs and PBMCs [8]. The NOD mutation affected the development of mouse macrophages and dendritic cells and the activity of mouse NK cells, which are known to be an important factor that impairs human engraftment [9,10]. In the early 2000s, the introduction of the IL-2 receptor common gamma chain deletion (*IL2Rγ^null^*) in the NOD-SCID background marked the next level of mouse strains available for humanization, generating the NSG mouse strain. The absence of *IL2Rγ* in these mice, which is a receptor for numerous mouse cytokines, impairs the development and function of B-, T- and NK cells [11,12,13]. Later on, the *IL2Rγ* deletion was combined with the NOD mutation and the deletion of the activating gene (Rag) 1 or 2 (*Rag1^null^* or *Rag2^null^*) [14] to generate the NRG mouse strain [15]. The deletion of the *Rag* genes makes mice resistant to irradiation, and therefore, they can be used for studies in cancer research in which irradiation and chemotherapy treatments are assayed. Apart from that, both NSG and NRG mouse strains have similar characteristics, and they both allow the engraftment of human HSC to a similar level [15]. For optimal human HSC engraftment, all immunodeficient mouse strains carrying the *IL2rγ^null^* deletion must be conditioned by sublethal radiation before transplantation. The level of irradiation applied differs depending on whether the host is a newborn or an adult mouse [16], and, as described above, whether it carries the *Rag* deletion.

To date, three major strains of immunodeficient mice are available to generate human immune system mice for their use in virology and immunology research: *NOD.Cg−Rag1^tm1Mom^IL−2Rγ_c_^tm1Wjlc^* (NRG) mice, *NOD.Cg-Prkdc^scid^Il2rg^tm1Sug^* (NOG) mice [11,17], and *NOD.Cg-Prkdc^scid^Il2rg^tm1Wjl^* (NSG) mice [4,5]. Furthermore, in recent years, NSG/NRG mice expressing human cytokines, human growth factors, or specific human HLA molecules have been developed. Of note are NSG-A2 mice, which express the human HLA class I-A2 molecule. By transplantation of HLA-A2-matching donor cells, these mice develop HLA-A2-restricted mature human T cells while maintaining the same number of total T cells [18,19]. Conversely, NRG-A2 mice transplanted with HLA-A2-matching donor cells show not only the development of HLA-A2-restricted mature T cells but also a considerably higher CD4^+^/CD8^+^ T cell count than classical NRG mice [20].

The most common methods to generate a human immune system in immunodeficient recipient mice include the transplantation of human peripheral blood lymphocytes (PBL model), hematopoietic stem cells (HSC model), and bone marrow-liver-thymus fetal tissue (BLT model) [2,21,22]. In the PBL model, human peripheral blood leukocytes (PBL) are transplanted into different mouse strains, resulting in rapid engraftment of human T cells, particularly CD3^+^ T cells. The main feature of this model is that the generated humanized mouse retains the immune memory of the donor cells. However, due to the presence of those mature T cells, these mice develop xenogeneic graft-versus-host disease (GvHD) within 16 weeks after transplantation [23], which narrows the window for their experimental use in long-term experiments [23]. Next-generation NSG strains, which are deficient in mouse MHC class I or II molecules, may extend such an experimental window by delaying the appearance of GvHD [11,24]. In the BLT model, small pieces of the human fetal liver and thymus are transplanted under the renal capsule, and CD34^+^ HSCs isolated from the same donor’s fetal liver are injected intravenously. BLT mice develop all human lineages, including mature HLA-restricted T cells. However, similar to the PBL model, they offer a limited window for experimental use due to the development of GvHD-like symptoms at around 20 weeks after transplantation [16,25]. In the HSC model, a complete human immune system can be established by transplantation of CD34^+^ hematopoietic stem cells (HSCs) of different origins (derived from bone marrow, cord blood, fetal liver, or mobilized peripheral blood). These mice are able to engraft physiological levels of human B cells, T cells and myeloid cells, including antigen-presenting cells. However, the number of human granulocytes, platelets, and red blood cells is usually low or absent, unless specific mouse strains are used that promote the development of these cell lineages, such as the NSGW41 mouse strain [26,27].

The complexity and diversity of available humanized mouse models and their use in different research areas has grown tremendously in recent years, giving the possibility to choose among a vast variety of models depending on the scientific question to be investigated [28,29]. Preclinical studies in virology, immunology and oncology have been conducted using some of these mouse models [2,30,31]. This review provides a general overview of the different protocols used to generate humanized mice, particularly NSG-HSCs and NRG-HSCs models, and their use in virology and immunology research in Germany. Table 1 summarizes the main aspects reviewed in this paper.

## 2. Humanization Protocols

Regardless of the mouse’s genetic background, the quality and quantity of human immune reconstitution can be significantly influenced by different factors, such as the source of human cells or tissues, the age and the sex of the mice, the transplantation route, and whether the mice are preconditioned before transplantation. These different factors have defined different protocols for humanization. In this review, we focused on general protocols involving humanized NRG and NSG mice for virology and immunology research (Figure 1) [40,41].

### 2.1. Generation of Humanized Mice by Transplantation in Young NSG Mice

Four- to six-week-old NSG or NSG-A2 mice are sub-lethally irradiated (i.e., 240 cGy). Four to twelve hours later, they are transplanted intravenously (tail vein or retro-orbital) with 7 × 10^5^ to 10^6^ CD34^+^ human hematopoietic stem cells (HSCs) per mouse, which are isolated from cord blood [35,37]. Eight to twelve weeks after transplantation, engraftment is confirmed in peripheral blood by detecting the number of human CD45^+^ cells by flow cytometry analysis. In addition to human and mouse CD45^+^ cells, a panel of specific blood cell markers can be used, including markers for lymphocytes (T, B), NK cells, and myeloid cells (monocytes, granulocytes, and dendritic cells). The percentage of engrafted human cells in peripheral blood depends on the quality and the number of transplanted HSCs. In infection studies, mice with an engraftment level between 10% to 60% are usually used [35,36,37,38,42]. These percentages refer to the number of human CD45^+^ cells within the total population of alive peripheral blood mononuclear cells (PBMCs).

### 2.2. Generation of Humanized Mice by Transplantation of Newborn NSG Mice

Newborn NSG or NSG-A2 mice are sub-lethally irradiated with lower irradiation doses (i.e., 100 cGy). Twenty-four to forty-eight hours later, mice are transplanted by intrahepatic injection with 3 × 10^5^ purified CD34^+^ HSCs from cord blood. Six weeks after transplantation, engraftment levels are evaluated in peripheral blood using flow cytometry, similar to the protocol used in young NSG mice [43,44,45,46]. The liver-mediated perinatal hematopoiesis significantly expands the hematolymphoid system during the first weeks of life in these animals, allowing the use of fewer HSCs to repopulate the immune system in newborn mice. This protocol showed better engraftment, expansion, and reconstitution of a human immune system, including T- and B-cells, compared to other protocols [47].

### 2.3. Generation of Humanized NRG Mice

Four- to five-week-old NRG mice are sub-lethally irradiated and four hours later they are transplanted with 2 × 10^5^ human CD34^+^ HSCs isolated from cord blood. Transplantation is performed intravenously through the tail vein. Since this mouse strain is more resistant to irradiation than NSG strains, mice can be irradiated with higher doses (i.e., 450 cGy). Frequencies of human immune cell subsets in peripheral blood are determined around 15 to 17 weeks after cell transplantation using flow cytometry and antibody panels similar to the protocols used in NSG mice [48,49,50,51,52].

## 3. Viral Infections in NSG and NRG Humanized Mouse Models

Different humanized mouse models have been used to study human-specific viral infections. In this section, we focus on NSG and NRG models developed in Germany to study the pathogenesis and immune responses to different human viruses.

### 3.1. Human Adenovirus Infection

Human adenovirus (HAdV) is a DNA tumor virus that usually causes asymptomatic to mild upper respiratory tract infections in children and young adults [53]. However, in immune-compromised patients, HAdV infection can cause severe disease characterized by the development of hepatitis, nephritis, and pneumonia, with possible lethal disease progression [54,55]. It is thought that these severe infections arise from the reactivation of a persistent infection due to immunosuppression [56].

The classical animal model to study HAdV infections has been the Syrian hamster model. This model has been mainly used to study the tumorigenic potential of some HAdVs, giving excellent comparability results between different studies [57]. However, Syrian hamsters infected with HAdVs do not exhibit human-like symptoms of acute respiratory infection or severe disease (i.e., viral hepatitis) and, therefore, severely limits its use to understand immune response and pathogenesis. Tree shrews are another promising animal model for HAdVs research [58]. In recent studies, it has been shown that Tree shrews develop human-like symptoms after HAdV infection. The limitation of this model is the lack of available molecular tools, the need for additional training to work with this model, and special requirements for maintenance [59].

As a model for human adenoviral infections, human immune system (HIS) NSG-A2 mice have been shown to be useful tools. Infection of HIS NSG-A2 mice with HAdV type 2 caused acute infection in one-third of infected mice, which showed weight loss, lethargy, ruffled fur, and death [37]. These acutely infected animals displayed visible signs of liver pathology, which agrees with a clinical picture of a severe adenoviral infection similar to the one observed in immunosuppressed patients [60]. Interestingly, mice with severe disease had high human chimerism, while animals with lower engraftment developed milder clinical symptoms or were asymptomatic. These results suggested that the level of engraftment is directly related to the severity of the disease, highlighting the importance of the presence of human immune cells to develop viral pathogenesis [37]. Interestingly, viral gene expression was still found in the bone marrow of asymptomatic mice long after infection, which would suggest the presence of persistent infection and would support the idea that severe HAdV infection in an immunocompromised patient could arise from the reactivation of persistent infection [61,62]. Moreover, these mice developed HAdV-specific immune responses shown by the production of IgM and antigen-specific CD8^+^ T cells. 

This model opens up new possibilities to understand adenovirus pathogenicity in a human-like environment. Future investigations in this model with other human adenovirus types could serve to understand the pathogenesis of this family of viruses in humans and could be used as a platform for the efficacy assessment of anti-viral drugs.

### 3.2. Ebola Virus Infection

The *Ebolavirus* genus encompasses six species of RNA viruses: Zaire, Sudan, Bundibugyo, Reston, Tai Forest, and Bombali, which cause different levels of pathogenicity in humans, i.e., from the highly pathogenic Zaire Ebola virus (EBOV), which causes case fatality rates of 90%, to the non-pathogenic Reston virus (RESTV), which does not cause disease in humans [63]. In the last decades, EBOV has caused multiple outbreaks in African countries that vary in the number of cases and their geographical extension. One of the most severe ones occurred in 2014–2016, which affected more than 28,000 people and caused more than 10,000 deaths. It spread to different parts of the African continent and reached Europe and America, causing an unprecedented global epidemic [64]. Interestingly, RESTV has only been shown to cause clinical illness in macaques and domestic animals such as pigs [65] and has been so far only associated with rare asymptomatic human infections [66].

Ebolavirus pathogenesis has been studied in non-human primates (NHPs), particularly in rhesus and cynomolgus macaques. As opposed to what happens in humans, NHPs are highly susceptible to all *Ebolavirus* species and can reach case fatality rates of about 90% [67,68]. On the contrary, ebolaviruses cannot infect wild-type mice, and although they can infect some immunocompromised mouse strains, they do not reproduce the clinical features of the Ebola virus disease observed in humans. For these reasons, they cannot be used to model the human pathogenesis of Ebola virus infection [69]. Different studies have established the use of HIS mice to model and understand the pathogenesis of these filoviruses in vivo.

NSG-A2-HSC mice have been successfully used to establish a model in which the pathogenesis of EBOV in humans can be reproduced. In this model, infection with EBOV causes weight loss, a decrease in survival, viremia, liver damage, splenomegaly, and hemorrhage [35]. NSG-A2-HSC mice have also been used to study the pathogenicity of the different *Ebolavirus* species in a human-like environment [38]. In those studies, HIS NSG-A2 mice intranasally infected with Zaire, Sudan, Bundibugyo, and Tai Forest *Ebolavirus* species recapitulated the case fatality rate observed in humans. Interestingly, RESTV infection showed 20% lethality in this mouse model, which was not expected. RESTV-infected mice that died seemed to have more virus replication in the liver than RESTV-infected survivor mice, suggesting that, although no RESTV disease has been yet detected in humans, this virus has the potential of being pathogenic if the mucosa immunity is overcome. 

Although this model has limitations in some functional aspects of the immune response, such as the lack of IgG class-switch, a poor CD4^+^ helper T cell response and it has underdeveloped lymph nodes, it could be a useful tool to understand the zoonotic potential of newly emerging filoviruses, their pathogenesis in humans and could serve as a platform for antiviral drug testing [38]. In summary, they could be an alternative to the use of NHPs since mice are easier to handle, cheaper to maintain, and do not require special animal facility infrastructures.

### 3.3. Dengue Virus (DENV) Infection

Dengue virus (DENV) belongs to the *Flaviviridae* family of RNA viruses. DENV is a mosquito-borne virus (also called arbovirus, for arthropod-borne virus) that is endemic in hundreds of tropical and subtropical countries. Approximately 400 million cases of dengue virus (DENV) infection are reported every year worldwide. [70]. In humans that suffer the infection for the first time, DENV causes asymptomatic to mild dengue fever disease. Subsequent DENV infections can cause more severe diseases, such as dengue hemorrhagic fever (DHF) and dengue shock syndrome (DSS), which can be fatal if not treated [71,72]. 

Different mouse models have been used to study DENV pathogenesis and to test antiviral drugs and vaccines against this virus. In this regard, the biggest challenge is still to perform studies on the immune response to DENV in a human context, which would help to better understand pathogenesis in humans, especially the severe forms of dengue. Having a model that can reproduce severe DENV infection would help to advance the design of new therapies. In the last few years, humanized mice have been established to study DENV infection in a human-like environment [44]. 

HIS NSG mice infected with DENV developed severe fever and erythema, and a decrease in the number of platelets in peripheral blood [43,46]. Moreover, different human immune cells in the spleen, the bone marrow, and blood were infected by DENV. Furthermore, high levels of cytokines and chemokines associated with severe DENV disease were found in infected humanized NSG mice [46]. Although these experiments showed some of the features of DENV infection in humans, the model could not recapitulate the pathogenesis of the severe forms of the infection, such as the hemorrhage that is observed in DHF or DSS patients [43,45]. Other studies have used HIS NSG mice injected with plasmids encoding IL-15 and Flt3L to promote NK cell development in this model and study the role of NK cells in the immune response to DENV infection. The presence of NK cells in these mice caused a subsequent reduction of virus replication, accompanied by thrombocytopenia and reduced liver damage. Furthermore, NK cells activated by infected monocyte-derived dendritic cells prevented DENV infection of monocytes, which was mediated by different adhesion molecules such as 2β4, LFA-1, DNAM-1, and CD2 [73].

Although studies of DENV infection in different mouse models, including humanized mice, have identified key viral and host factors, further studies will help to improve our understanding of host-virus interactions and, therefore, will lead to the development of efficient vaccines and antiviral drugs.

### 3.4. Human Cytomegalovirus (HCMV) Infection

More than 90% of adults in developing countries and 60% of adults in developed countries have been infected with human cytomegalovirus (HCMV) in their lifetime [74]. HCMV is a herpesvirus that causes asymptomatic to mild disease and induces a robust immune response. After suffering an acute infection, HCMV persists in immune cells for life, causing a latent infection. During episodes of immunosuppression, HCMV can be reactivated, causing more severe infections in those patients. Moreover, HCMV can cause severe congenital infection in newborns and has also been associated with tumor development and neurological disorders [75,76]. HCMV latency, reactivation, and pathogenesis in immunocompromised individuals have been extensively investigated [77,78,79]. Many studies have used murine (MCMV) or rhesus monkey CMV (RhCMV) viruses to understand the pathogenesis of HCMV in animal models. However, the immune response and the clinical features of MCMV and RhCMV infection in those animal models are very different from the ones observed in humans infected with HCMV [80,81]. This has limited the preclinical evaluation of human vaccines [82].

Several studies have used HIS NRG mice to model HCMV infection and immune response [48,51,52,82]. In this model, HCMV reactivation by treatment with recombinant human G-CSF promoted the detection of the virus in the spleen, lymph nodes, liver, salivary glands and bone marrow. Interestingly, HCMV infection enhanced T cell development in the thymus of these mice, which was associated with an expansion of memory CD4^+^ and CD8^+^ T cells in secondary lymphatic tissues and upregulation of the PD1 activation marker. In addition, due to the infection and reactivation, HIS NRG mice showed increased follicular helper T cells. Plasma samples were found to contain HCMV-specific IgM and IgG antibodies [48].

HIS NRG mice have also been used as a platform to create monoclonal IgGs antibodies that can be used for passive immunization against HCMV. Vaccination of HIS NRG mice with dendritic cells expressing GM-CSF, interferon alpha, and an HCMV antigen (gB) generated different anti-gB IgG antibodies. Two of these antibodies, which were selected and synthetically produced, showed high neutralization efficacy in vivo and full protection against HCMV challenge in HIS NRG mice. [51].

In another study, HIS NSG-A2 mice were used to test a live-attenuated vaccine against HCMV infection [82]. This vaccine was based on dendritic cells that were loaded with HCMV antigens by infection with an attenuated HCMV strain in vitro.. Three months after vaccination with HCMV-loaded dendritic cells, mice were challenged with the wild type virus and a week later a booster of the vaccine was administered. In these experiments, mice showed to be fully protected against HCMV. In addition, and similar to the HIS NRG model, HCMV infection induced virus-specific CD8^+^ T cells and IgM responses in HIS NSG-A2 mice [82,83].

Further optimization of currently available humanized mouse models will serve to further understand the complexity of the immune response generated upon HCMV infection and reactivation. 

### 3.5. Human Immunodeficiency Virus (HIV) Infection

Currently, approximately 40 million people worldwide live with HIV, and more than 600,000 people died in 2021 of HIV-related illnesses [84,85]. Even though there is still no cure for HIV infection, the treatment with antiretroviral therapy (ART) has turned HIV deadly infection into a chronic disease. However, being chronically infected with HIV increases the risk of suffering from other diseases such as heart, bone, liver, kidney, and neurological diseases, as well as other infectious diseases [86]. Although many advances have been made in the last 40 years, there are some aspects of HIV infection that need further research, such as the relapse during ART and the design of new therapeutic approaches that would serve to eliminate the virus for life. 

Classically, non-human primates (NHPs) have been used as the primary model for HIV-1 research, which has provided crucial insights into viral pathogenicity [87]. However, financial limitations and a lack of appropriate facilities have severely limited the use of this model. HIV infection was one of the first infections modeled in humanized mice. HIS mice made in the NOG [23], NSG [88,89], and NRG-A2 backgrounds [90], as well as the BLT model in NSG mice [88,91], and many others [92], have been extensively used to study HIV pathogenesis and anti-viral therapy.

A recent study using HIS NRG mice revealed that type 1 interferon (IFN) signaling persisted after antiretroviral therapy and explored the role of dendritic cells during HIV-1 infection and latency [93]. In addition, broadly neutralizing antibodies and other therapeutics that enhance ART treatment have been shown to prevent HIV-1 transmission from cell to cell in this model [94]. Furthermore, humanized mice with chronic HIV-1 infection that were treated with combination antiretroviral therapy (cART) developed hepatitis and liver fibrosis, which is a common comorbidity observed in chronically infected ART-treated human patients. In these animals, hepatitis was associated with the accumulation of M2-like macrophages and the elevation of TGF-β and IFN signaling in the liver. Accordingly, the inhibition of IFN-I signaling reversed HIV/cART-induced liver disease in humanized mice [95].

HIS NSG mice have also been infected with HIV, showing efficient virus replication and depletion of CD4^+^ T cells. Additionally, a diffuse distribution of CD4^+^ T cells and CD68^+^ macrophages was observed in the cervical-vaginal area, suggesting that this model could support HIV infection through vaginal exposure. Furthermore, cART treatment efficiently inhibited the exponential growth of the virus and reduced plasma viral RNA to undetectable levels within four weeks post-treatment initiation [88]. Plasma viremia levels of patients taking cART often rebound rapidly following a treatment interruption, reaching levels comparable to the ones detected before therapy began [96]. This key aspect of HIV infection in humans was recreated in this mouse model after the discontinuation of cART therapy. HIV levels rebounded and peripheral CD4^+^ T cells decreased rapidly once cART therapy stopped. These results showed that HIS NSG mice can recapitulate HIV infection and pathogenesis in vivo [88].

Other studies have used HIS NSG mice to investigate the efficacy of LASER ART (long-acting slow-effective release antiviral therapy) and CRISPR-Cas9 combination therapy to excise HIV-1 proviral DNA fragments integrated into the genome of latently infected cells. Consequently, HIV-1 subgenomic DNA was eliminated in vivo and could not be further detected in blood, lymphoid tissue, bone marrow, or brain. These results showed the possibility of permanent virus elimination in human patients [97,98].

The use of humanized mouse models has made HIV-1 research more cost-effective and more accessible to scientists around the world. Currently, NRG mice and other humanized mouse models have become a useful preclinical tool to examine the direct interaction between HIV-1 and the human immune system, and have allowed investigations on more complex clinical problems such as the HIV/cART-associated hepatitis, the persistence of HIV infections, and the immune response to co-infections with other microorganisms such as *Mycobacterium tuberculosis* [99].

### 3.6. Hantaan Orthohantavirus (HNTV) Infection

Hantaviruses, particularly Hantaan orthohantavirus (HNTV), are zoonotic pathogens of public health importance, which are found in all continents except Antarctica. These viruses can cause different diseases in humans, namely hemorrhagic fever with renal syndrome (HFRS), Hantavirus pulmonary syndrome (HPS), or Hantavirus cardiopulmonary syndrome (HCPS) [100]. These disorders are associated with high fever, acute thrombocytopenia and changes in vascular permeability, and with renal and/or pulmonary symptoms [101]. Multiple animal models, such as mice, rats, hamsters, and NHPs, have been used to study pathogenesis and immune response to HNTV infection. Most of these models partially resemble clinical signs of the diseases observed in humans. However, better models are needed to understand the immune response to HNTV in humans [102].

HNTV pathogenesis has been studied in HIS NSG and HIS NSG-A2 mice [36]. In both models, HNTV infection caused clinical signs of disease; however, higher lethality and more severe symptoms were observed in HIS NSG-A2 mice. Additionally, HNTV RNA was found in different organs, such as the kidneys, liver, spleen, and lungs. Furthermore, HIS NSG-A2 mice lost weight faster, had a higher amount of infiltrated inflammatory cells in their lungs, and a higher reduction in the number of platelets in their blood than HIS NSG mice [36,103]. This reduction in platelet numbers is known as one of the hallmarks of Hantavirus disease in humans [104]. Interestingly, this phenotype was not observed in non-humanized Hantavirus-infected NSG mice, suggesting that the presence of the human immune cells was related to pathogenesis.

Another study in HIS NSG-A2 mice showed a correlation between productive infection in monocytes and dendritic cells, and the pathogenicity of different Hantaviruses [42]. Moreover, other studies showed that the high viral load observed in the lungs of infected humanized mice was mainly coming from immune cells expressing human MHC class II and β2 microglobulin molecules. Infection of immune cells in the lungs caused capillary leakage and subsequent non-cardiogenic acute pulmonary edema, which is one of the syndromes observed in human patients [42,101] These results suggested that the pathogenicity of the different *Hantavirus* species in humanized mice depends on their replication capacity in immune cells and on how this replication influences the functionality of these cells.

In summary, humanized mice recapitulate some key aspects of Hantavirus disease in humans, such as the high amount of immune cell infiltration in the lungs and the reduced platelet number. These studies described for the first time that humanized mice can serve as a good model to study human Hantavirus immunopathogenesis and, therefore, it could be used to assess the virulence of newly emerging Hantaviruses in humans and to test the efficacy of antiviral drugs against them.

## 4. Closing Remarks and Future Perspectives

Humanized mice have become a powerful research tool for studying human-specific viral infectious diseases. A significant improvement has been made in recent years to generate new immunodeficient mouse strains and new strains expressing human growth factors and cytokines. These new strains allow higher degrees of human immune chimerism, reduce the development of side effects, such as GvHD, and stabilize the levels of engrafted cells over time. These new models have increased the use of humanized mice to better understand not only the pathogenesis of classical human-specific viruses but also of emerging and re-emerging viruses. These models have served to understand the contribution of the human immune system to the pathogenesis observed upon infection and have been postulated as a useful tool to investigate the zoonotic potential of newly appearing viruses. Moreover, humanized mice have been widely used for in vivo validation of new therapeutic approaches, including drugs, vaccines, and immunotherapy, establishing them as a new gold standard model in translational research [11,105,106,107,108,109].

In comparison to other models that are used to study human-specific diseases such as non-human primates (NHPs), humanized mouse models are cheaper and easier to maintain, easier to handle, and experiments with them can be performed in any lab with a mouse facility. Furthermore, humanized mice have been shown, at least for some viruses, to more closely recapitulate the pathogenesis of human diseases, in vivo.

Despite all of the progress achieved in recent years, several challenges lie ahead to further improve these models. More mouse models that are able to develop functional lymph nodes [110], IgM to IgG class-switch, the development of models that combine both human immune system and human tissues (lungs [111], liver [112], brain [113], and pancreas [114]) will open up new opportunities to understand, for example, the pathogenesis of respiratory viruses, liver-tropic viruses, or brain-tropic viruses in a human-like environment. Further reduction of the host’s innate immunity and new humanization protocols in which transplantation of immune cells and tissues are combined, need to be developed.

## Figures and Tables

**Figure 1 viruses-15-00478-f001:**
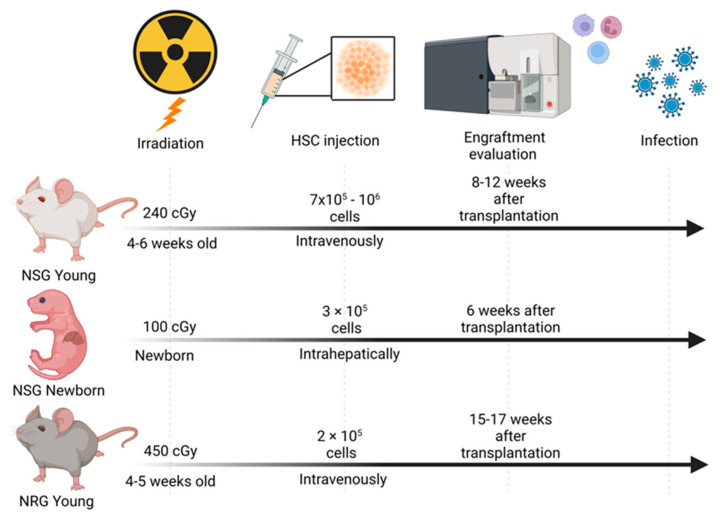
Schematic representation of the different steps to generate humanized NSG and NRG mice for virology and immunology research. Created with BioRender.com.

**Table 1 viruses-15-00478-t001:** Summary of the humanized mouse models and applications reviewed in this paper.

Year of Generation	Virus	Mouse Strain	Humanization	Reference
2010 2011	HIV	NSG NRG	CD34^+^, newborn	[32,33]
2014	HCMV	NSG NSG-A2	CD34^+^, newborn	[34]
2015	EBOV	NSG-A2	CD34^+^ HSC, young mice	[35]
2015	Hantaviruses	NSG	CD34^+^ HSC, young mice	[36]
2017	HAdV	NSG-A2	CD34^+^ HSC, young mice	[37]
2019	EBOV, SUDV, RESTV, etc.	NSG-A2	CD34^+^ HSC, young mice	[38,39]

## Data Availability

Not applicable.

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
