# Peer review of "Recent Developments in NSG and NRG Humanized Mouse Models for Their Use in Viral and Immune Research"

_viruses, 2023, doi:10.3390/v15020478_

Round 1

Reviewer 1 Report

It is a very well-written, comprehensive, and concise review.  The topic of the review is also very interesting and relevant to biomedical research. The review in its present form is acceptable for publishing. 

In this review, Kitsera et al., summarized the recent developments in the field of animal models with a specific focus on the generation of humanized mouse models which can be used to study viral infections in humans. The review elegantly summarised the history of the development of immunocompromised mouse models. The current review mainly focuses on the development and the specific use of the NSG and NRG mice models in the context of the successful engrafting of hematopoietic stem cells. The topic chosen is vital for the pathophysiological understanding of human infectious diseases from a clinical point of view. As the science behind the development of these mouse models and the possible benefits are seldom understood completely by clinicians. The current review discussed different protocols used in generating humanized mice. This is important as the specific protocol selection can significantly impact the humanized immune microenvironment in these mouse models regardless of the genetic profile of the mouse. The specific use of these humanized mice models in HAdVs, EBOVs, DENVs, HCMVs, HIVs, and HNTVs infections was discussed with a focus on the human similarity of the clinical signs exhibited by these mouse models. However, the review did not elaborate on the use of these mouse models for the therapeutic validation of different potential drugs. The review also did not highlight the potential limitations and/or advantages posed by these rodent models as compared to non-human primate models (NHPs) in preclinical studies. A concise table should be added that enlists the different humanized mouse models developed over time in chronological order along with their specific protocols.

Author Response

Dear Editor and Reviewers,

We thank you for the revision of our manuscript.

In this letter we address the comments and suggestions given by the reviewers. We have done a deep review of the stile and English grammar without altering the content or structure of the text. We hope that the manuscript is now ready for publication in Viruses.

Comments Reviewer 1:

In this review, Kitsera et al., summarized the recent developments in the field of animal models with a specific focus on the generation of humanized mouse models which can be used to study viral infections in humans. The review elegantly summarised the history of the development of immunocompromised mouse models. The current review mainly focuses on the development and the specific use of the NSG and NRG mice models in the context of the successful engrafting of hematopoietic stem cells. The topic chosen is vital for the pathophysiological understanding of human infectious diseases from a clinical point of view. As the science behind the development of these mouse models and the possible benefits are seldom understood completely by clinicians. The current review discussed different protocols used in generating humanized mice. This is important as the specific protocol selection can significantly impact the humanized immune microenvironment in these mouse models regardless of the genetic profile of the mouse. The specific use of these humanized mice models in HAdVs, EBOVs, DENVs, HCMVs, HIVs, and HNTVs infections was discussed with a focus on the human similarity of the clinical signs exhibited by these mouse models. However, the review did not elaborate on the use of these mouse models for the therapeutic validation of different potential drugs. The review also did not highlight the potential limitations and/or advantages posed by these rodent models as compared to non-human primate models (NHPs) in preclinical studies. A concise table should be added that enlists the different humanized mouse models developed over time in chronological order, along with their specific protocols.

Response: We thank the reviewer for the suggestion on doing a more elaborate review of the use of humanized mice for therapeutic validation of different drugs. We think this is an interesting topic; however, it is out of the scope of this review, which intends to summarize the available humanized mouse models, the protocols to generate them and the viral diseases for which they have been used. Nevertheless, the review slightly mentions the use of humanized mice in therapeutics. The use of humanized mice for therapeutic validation would be an interesting topic for another review paper.

With regard to the comparison between NHPs and humanized mice advantages and disadvantages, some sentences have been now added in Lines 542

A table has been now added to summarize the models, the humanization protocols and their uses in virology and immunology research (Line 132).

Sincerely,

Dr. Maksym Kitsera (on behalf of all authors).

Reviewer 2 Report

In this review, Kitsera and colleagues summarize studies that used humanized mouse models to investigate pathologies of human viral infections. After describing the currently available models, they provide selected examples of viral infections that took advantage of these models (human adenoviruses and HCMV, HIV, Dengue virus, ebolaviruses and hantaviruses. The review is well structured and documented. The selected examples are relevant. It is unclear why they were selected over others. In particular, EBV has long been tested in humanized mice. In the case of EBV, recent reviews may be cited or the absence of research on this virus in Germany may be mentioned to justify the absence of that virus from this review (it is not required from the authors to add a chapter).

The review provides sufficient coverage and detail of the topic. A few points, listed below, should be addressed to provide some clarification. The text would benefit from a thorough review of its English grammar to make it easier to read.

Points of clarification

1) L127: explain what the percentages refer to. For instance, what is the reference population?

2) L178-180: it seems paradoxical that HIS-reconstituted mice are needed to reproduce adenoviral pathologies seen in immunosuppressed humans, who do not have a fully functional immune system. Can the author elaborate whether the HIS mice have a fully functional human immune system, or an immune system that is more representative of that of an immunosuppressed human.

3) L208-210: not clear why susceptibility of NHPs to RESTV limits or disqualifies the NHP models for other ebolaviruses. Following that reasoning, the fact that humanized mice succumb to RESTV (Lines 222-223) would make them an unsuitable model for the study of ebolavirus pathologies.

4) L228: in some aspects. Please explain

5) L287-288: could the authors provide detail how HCMV reactivation is studied in these mice.  Is the model used with HCMV+ HPSC?

6) L408: claiming that humanized mice are considered “gold standard” in translational research needs to be supported by citations and references.

7) Grammar and typos (not a complete list):

L22: immune system (mice have only one immune system) if the sentence refers to innate and adaptive immune systems, it should be specified.

L44-46: sentence needs clarification (use of “and”)

L59: radioresistance

L61: experimental settings such as

L86: (REF) reference missing

L88, 92, etc: abbreviation GvHD or GVHD must be consistent

L88: “the human fetal liver and thymus are transplanted” indicates that the whole organs are transplanted. Needs clarification.

L103: in virology

Legend figure 1 (and L114): the process described is not the process of generating NRG and NSG mice. These mice have been generated genetically. What is described is the generation of humanized NRG and NSC mice for virology and immunology research. Please clarify throughout the manuscript.

L421: Schematic representation

L142: weeks old (no hyphen in this case)

L144: performed intravenously

L173: requirements; or “a special requirement”, in which case, this single requirement must be specified

L198-200: remove “specie”; the mentions of Ebola virus and Reston virus are sufficient as definitions

L257: reduction of

L260 of monocytes

L261: studies of

L269: human cytomegalovirus (or possibly Human Cytomegalovirus)

L271: after causing

L280: rephrase: thereby limiting the preclinical evaluation of human vaccines

L293: reference needed for the study

L306: replace “last year” with actual date (did you mean 2022 or 2021?)

L385: The authors of that study hypothesized

Author Response

Dear Editor and Reviewers,

We thank you for the revision of our manuscript.

In this letter we address the comments and suggestions given by the reviewers. We have done a deep review of the stile and English grammar without altering the content or structure of the text. We hope that the manuscript is now ready for publication in Viruses.

Comments Reviewer 2:

In this review, Kitsera and colleagues summarize studies that used humanized mouse models to investigate pathologies of human viral infections. After describing the currently available models, they provide selected examples of viral infections that took advantage of these models (human adenoviruses and HCMV, HIV, Dengue virus, ebolaviruses and hantaviruses. The review is well structured and documented. The selected examples are relevant. It is unclear why they were selected over others. In particular, EBV has long been tested in humanized mice. In the case of EBV, recent reviews may be cited or the absence of research on this virus in Germany may be mentioned to justify the absence of that virus from this review (it is not required from the authors to add a chapter).

Response: We agree with the reviewer, humanized mice have been extensively used to study EBV infection and therapeutics. However, this virus was not included in our list because the research has not been performed in Germany. Most of the studies have been done by Prof. Christian Münz, who is located at University of Zürich, in Switzerland.

The review provides sufficient coverage and detail of the topic. A few points, listed below, should be addressed to provide some clarification. The text would benefit from a thorough review of its English grammar to make it easier to read.

Response: We have now reviewed the English stile and grammar throughout the manuscript.

Points of clarification

1) L127: explain what the percentages refer to. For instance, what is the reference population?

Response: This has been now added (line 170)

2) L178-180: it seems paradoxical that HIS-reconstituted mice are needed to reproduce adenoviral pathologies seen in immunosuppressed humans, who do not have a fully functional immune system. Can the author elaborate whether the HIS mice have a fully functional human immune system, or an immune system that is more representative of that of an immunosuppressed human.

Response: The humanized mouse model used in the Adenovirus studies reconstitutes all main blood cell subpopulations, like B cells, T cells and Myeloid cells. However, it is known that the lymph nodes in this model are not well developed  and that the germinal centers at the spleen are also not well structured. These features prevent these animals from producing IgG immunoglobulin in response to infection, for example. The number of NK cells is also very low and no human erythrocytes are reconstituted.  So, this model lacks some functional aspects of a mature immune system and limits its use to study immune response to infection or vaccination, for which it would be more appropriate to use other models like the PBL model, for example. However, many studies have shown that it is a good model to study pathogenesis in vivo of human-specific pathogens.

3) L208-210: not clear why susceptibility of NHPs to RESTV limits or disqualifies the NHP models for other ebolaviruses. Following that reasoning, the fact that humanized mice succumb to RESTV (Lines 222-223) would make them an unsuitable model for the study of ebolavirus pathologies.

Response: These lines have now been rephrased for a better understanding.

4) L228: in some aspects. Please explain

Response: This has been now explained in the text (Line 298).

5) L287-288: could the authors provide detail how HCMV reactivation is studied in these mice.  Is the model used with HCMV+ HPSC? (added) This model does use it)

Response: An explanation of how the reactivation of HCMV was done has been added to the manuscript (Line 377)

6) L408: claiming that humanized mice are considered “gold standard” in translational research needs to be supported by citations and references. (removed).

Response: We have now added some references to this sentence.

7) Grammar and typos (not a complete list):

L22: immune system (mice have only one immune system) if the sentence refers to innate and adaptive immune systems, it should be specified.

L44-46: sentence needs clarification (use of “and”)

L59: radioresistance

L61: experimental settings such as

L86: (REF) reference missing

L88, 92, etc: abbreviation GvHD or GVHD must be consistent

L88: “the human fetal liver and thymus are transplanted” indicates that the whole organs are transplanted. Needs clarification.  

L103: in virology

Legend figure 1 (and L114): the process described is not the process of generating NRG and NSG mice. These mice have been generated genetically. What is described is the generation of humanized NRG and NSC mice for virology and immunology research. Please clarify throughout the manuscript.

L421: Schematic representation

L142: weeks old (no hyphen in this case)

L144: performed intravenously

L173: requirements; or “a special requirement”, in which case, this single requirement must be specified

L198-200: remove “specie”; the mentions of Ebola virus and Reston virus are sufficient as definitions

L257: reduction of

L260 of monocytes

L261: studies of

L269: human cytomegalovirus (or possibly Human Cytomegalovirus)

L271: after causing

L280: rephrase: thereby limiting the preclinical evaluation of human vaccines

L293: reference needed for the study

L306: replace “last year” with actual date (did you mean 2022 or 2021?)

L385: The authors of that study hypothesized

Response: Typos and grammar have been now corrected, and doubts have been clarified.

Sincerely,

Dr. Maksym Kitsera (on behalf of all authors).
